# Adaptive $k$-nearest neighbor classifier based on the local estimation of the shape operator

## Abstract

Nonparametric classification does not assume a particular functional form of the underlying data distribution, being a suitable approach for a wide variety of data sets. The $k$-nearest neighbor ($k$-NN) algorithm is one of the most popular methods for nonparametric classification. However, a relevant limitation concerns the definition of the number of neighbors $k$. This parameter exerts a direct impact on several properties of the classifier, such as the bias-variance tradeoff, smoothness of decision boundaries, robustness to noise, and class imbalance handling. In the present paper, we propose a new adaptive $k$-nearest neighbours ($kK$-NN) algorithm that explores the local curvature at a sample to automatically define the neighborhood size. The rationale is that points with low curvature could have larger neighborhoods (locally, the tangent space approximates well the underlying data shape), whereas points with high curvature could have smaller neighborhoods (locally, the tangent space is a loose approximation). We estimate the local Gaussian curvature by computing an approximation to the local shape operator in terms of the local covariance matrix as well as the local Hessian matrix. Results on many real-world data sets indicate that the new $kK$-NN algorithm may provide superior balanced accuracy compared to the established $k$-NN method. This is particularly evident when the number of samples in the training data is limited, suggesting that the $kK$-NN is capable of learning more discriminant functions with less data in some relevant cases.

## 1 Introduction

Supervised classification stands as a cornerstone in machine learning, providing a foundational concept to address a myriad of real-world problems. Its training process aims to recognize patterns within labeled data sets, where each data point is associated with a predefined class (Sen et al., 2020). The algorithm learns to predict the class of unseen instances based on their features, enabling automated decision-making and pattern recognition. From medical diagnosis to fraud detection to image recognition and sentiment analysis, supervised classification empowers data scientists and machine learning practitioners to extract valuable insights and make informed decisions from large and complex data sets (El Mrabet et al., 2021). Supervised learning algorithms are broadly divided into parametric and nonparametric methods, each offering distinct advantages and attributes (Nagdeote and Chiwande, 2020; Mahesh et al., 2023; Gramacki, 2018).

Parametric classifiers assume a specific functional form of the underlying data distribution, with a finite number of parameters remaining constant regardless of the size of the data set. Parametric classifiers such as logistic regression and Bayesian classifiers, are characterized by their simplicity and computational efficiency, being well-suited to scenarios with limited data and well-defined assumptions about the data distribution (Wen et al., 2023). Conversely, nonparametric classifiers - including $k$-nearest neighbors, decision trees, and support vector machines (SVM), do not make explicit assumptions about the underlying data distribution (Braga-Neto, 2020). Instead, they rely on flexible models that adapt to the complexity of the data, frequently requiring further computational resources and larger data sets while offering greater flexibility and robustness (Wang et al., 2023). Understanding the distinctions between parametric and nonparametric classifiers is crucial for selecting appropriate models that best capture the underlying structure of the data and fulfill the requirements of the specific problem domain.

Nonparametric classifiers provide a relevant alternative by providing flexibility, robustness, and adaptability to complex data sets without imposing strict assumptions on the underlying data distribution (Cai and Wei, 2021). Unlike parametric methods, nonparametric classifiers are not bound by predetermined functional forms, allowing them to effectively capture intricate patterns and relationships present in diverse data sets. This inherent flexibility enables nonparametric classifiers to handle non-linear and high-dimensional data, being well-suited for tasks where the underlying distribution is unknown or difficult to specify a priori (Modarres, 2023; Chen et al., 2023). In addition, nonparametric classifiers are particularly suitable to handle noisy data and outliers, due to the fact that they do not rely on strict assumptions on the data distribution (Trillos et al., 2019; Levada and Haddad, 2021; Sert and Kardiyen, 2023). Overall, the advantages of nonparametric classifiers lie in their ability to offer versatile and robust solutions for a wide range of machine learning tasks (Bawono et al., 2020; Ren and Mai, 2022).

The $k$-nearest neighbor classifier ($k$-NN) is a nonparametric method known for its simplicity, versatility, and intuitive approach to classification tasks (Cover and Hart, 1967; Nielsen, 2016). The $k$-NN algorithm is well-suited to handling complex, non-linear relationships and high-dimensional data sets where the underlying structure may be difficult to specify (Zhang, 2022). Another advantage is its ease of implementation and interpretability, being the classification decision determined by the majority class among the $k$-nearest neighbors of a given data point. Moreover, the $k$-NN classifier requires minimal training time since it essentially memorizes the training data set, becoming suitable for both online and offline learning scenarios. Furthermore, the $k$-NN algorithm adapts dynamically to changes within the data set, being more robust to noise and outliers (Syriopoulos et al., 2023).

In the $k$-NN classifier, the parameter $k$ controls the neighborhood size and it plays a crucial role in determining behavior and performance of the model. This parameter represents the number of nearest neighbors considered when making predictions for a new data point (Jodas et al., 2022). A smaller $k$ value leads to a more flexible model with decision boundaries that closely follow the training data, potentially capturing intricate patterns and local variations. However, smaller $k$ values may also increase the model's susceptibility to noise and outliers, as it excessively relies on the nearest neighbors for classification (Uddin et al., 2022). Conversely, larger $k$ values result in a smoother decision boundary and a more generalized model that is less affected by individual data points. However, exceedingly large $k$ values may cause the model to overlook local patterns, resulting in an inferior performance - notably in data sets with complex structures. Therefore, selecting the appropriate $k$ value is pivotal for achieving an appropriate balance between bias and variance, ensuring optimal generalization and predictive accuracy on testing or new data. The parameter $k$ is commonly the same for all samples in the data set (Batista and Silva, 2009).

The motivation of the present work is to improve the performance of the $k$-NN classifier through the incorporation of a geometric property (local curvature) in the definition of the neighborhood size. A high curvature sample $\vec{x}_i$ should have a smaller neighborhood, as the patch $P_i$ composed by $\vec{x}$ and its closest neighbors deviates from a linear subspace. Conversely, low curvature samples should have larger neighborhoods, as the patch $P_i$ is approximately linear. Several works have investigated the incorporation of other distance functions and adaptive ways to define the parameter $k$ automatically for each data set (Alfeilat et al., 2019; Zhao and Lai, 2021; Papanikolaou et al., 2021; Daulay et al., 2023). Techniques for the local adaptive estimation of the neighborhood size for each sample of a data set have been proposed for the $k$-NN classifier (Sun and Huang, 2010; Fan et al., 2023). Nonetheless, most of the literature adopts the optimization of a local criterion to choose the best value of $k$ from a list of candidates.

In the present paper, we introduce an adaptive curvature based $k$-nearest neighbor classifier to automatically adjust the number of neighbors for each sample of a data set. The proposed method is named $kK$-NN due to the fact that it consists of an $k$-NN method with $k$ varying locally according to the Gaussian curvature. The intuition behind the $kK$-NN classifier is to explore the local curvature to define the size of the neighborhood $k$ at each vertex of the $k$-NNG in an adaptive manner. As detailed in Algorithm 1, in the case of points with a lower curvature values, the tangent plane is commonly closely adjusted to a manifold. The $kK$-NN classifier is composed by the training and testing stages (algorithms 2 and 3, respectively). In the training stage, the first step consists in building the $k$-NNG from the input feature space using $k = log_2\ n$, where $n$ is the number of samples. Subsequently, it is computed the curvature of all vertices of the $k$-NNG exploring the shape operator-based algorithm. The curvatures are then quantized into ten different scores. Considering

that the scores are based on the local curvatures, the adaptive neighborhood adjustment is performed by pruning the edges of the $k$-NNG.

There are three main contributions of the proposed $kK$-NN algorithm. Firstly, transforming the number of neighbors spatially-invariant along a $k$-NN graph in order to avoid both under and over fitting. Secondly, to control the smoothness of the decision boundaries, depending on the local density in the feature space. Thirdly, to improve the robustness to noise and outliers through the identification of high curvature points adopting an approximation for the local shape operator of the data manifold (do Carmo, 2017; Needham, 2021; Boissonnat and Wintraecken, 2022). Computational experiments with 30 real-world data sets indicate that the proposed $kK$-NN classifier is capable of improving the balance accuracy compared to the existing $k$-NN, especially while dealing with small training data sets.

The results indicate that the $kK$-NN classifier is consistently superior compared to the regular $k$-NN. The rationale behind the capacity of the $kK$-NN to improve the regular $k$k-NN may be summarized by three relevant aspects. Firstly, through the use of an adaptive strategy to define the neighborhood sizes, the $kK$-NN is capable of avoiding either underfitting or overfitting. Secondly, in more dense regions the decision boundaries becomes more adjusted to the samples, while in less dense regions the boundaries become smoother - making the classification rule adaptive to different regions of the feature space. Thirdly, high curvature points are commonly related to outliers, being the $kK$-NN classifier capable of isolating such samples by drastically reducing its neighborhoods. It is worth mentioning that the $kK$-NN classifier is capable of learning more discriminant decision functions when the number of training samples is considerably reduced. These results suggest that the $kK$-NN algorithm may provide more flexible and adjustable decision boundaries, while reducing the influence of outliers over the classification process.

The remainder of the paper is organized as follows. Section 2 presents the proposed adaptive curvature-based $kK$-NN classifier. Section 3 reports computational experiments and results. Section 4 concludes and suggests future research possibilities.

## 2   Adaptive curvature based $k$-NN classifier

One of the main limitations of the $k$-NN classifier is related to parameter sensitivity, as the performance of the $k$-NN algorithm is highly dependent upon the choice of the parameter $k$ - i.e. the number of nearest neighbors to consider. Selecting an inappropriate value of $k$ may result in a series of negative effects to its performance such as (Zhang, 2022; Syriopoulos et al., 2023; Daulay et al., 2023):

- **Overfitting and underfitting:** The parameter $k$ controls the flexibility of the decision boundary in the $k$-NN classifier. A smaller value of $k$ results in a more flexible (less smooth) decision boundary, which may lead to overfitting - particularly in noisy or high variance data sets. Conversely, a larger value of $k$ results in a smoother decision boundary, which may lead to underfitting and inappropriate generalization in the case that the $k$ is too large compared to the data set size or the underlying structure of the data.

- **Bias-variance tradeoff:** The choice of $k$ in the $k$-NN classifier involves a trade-off between bias and variance. A smaller $k$ leads to low bias but high variance. This means that the classifier might capture more complex patterns in the data, although it is sensitive to noise and fluctuations. On the other hand, a larger $k$ reduces variance while increases bias, potentially leading to simpler decision boundaries that may not capture the true underlying structure of the data.

- **Robustness:** The sensitivity of the $k$-NN classifier to the $k$ parameter also affects its robustness to noisy data points and outliers. A larger $k$ may mitigate the effects of noise by considering a larger number of neighbors, whereas a smaller $k$ may lead to overfitting - in which case the classifier is more affected by noisy data points.

- **Impact on class-imbalanced data sets:** In class-imbalanced data sets, the minority class (i.e. a class with fewer instances) tends to be underrepresented compared to the majority class. The choice of the $k$ parameter may influence the classification of minority class instances. A smaller $k$

may result in the majority class dominating the prediction for the minority class instances. Since the nearest neighbors are predominantly from the majority class, the minority class instances might be misclassified or even ignored. A larger $k$ may incorporate more neighbors, potentially improving the representation of the minority class. However, it might also introduce noise from the majority class, leading to misclassification of minority class instances.

## 2.1 Algorithm to estimate the shape operator curvature

In order to propose our method to compute Gaussian curvatures, suppose that $X = \{\vec{x}_1, \vec{x}_2, \ldots, \vec{x}_n\}$, where $\vec{x}_i \in \mathbb{R}^m$, denotes the input of the data matrix, in which each column of $X$ represents a sample of the data set. Given matrix $X$, we may build a graph from the $k$-nearest neighbors of each sample ($k$-*nearest neighbor graph*), known as $k$-NNG. For each sample, calculate its $k$-nearest neighbors based on a distance metric (e.g., Euclidean distance, Manhattan distance, cosine similarity) in the feature space. Then, add an edge linking each sample to its $k$-nearest neighbors, creating a graph where each sample is a node, and edges represent the connections between a pair of nodes (Eppstein et al., 1997). Figure 1 illustrates an example of an $k$-NNG created using a real-world data set.

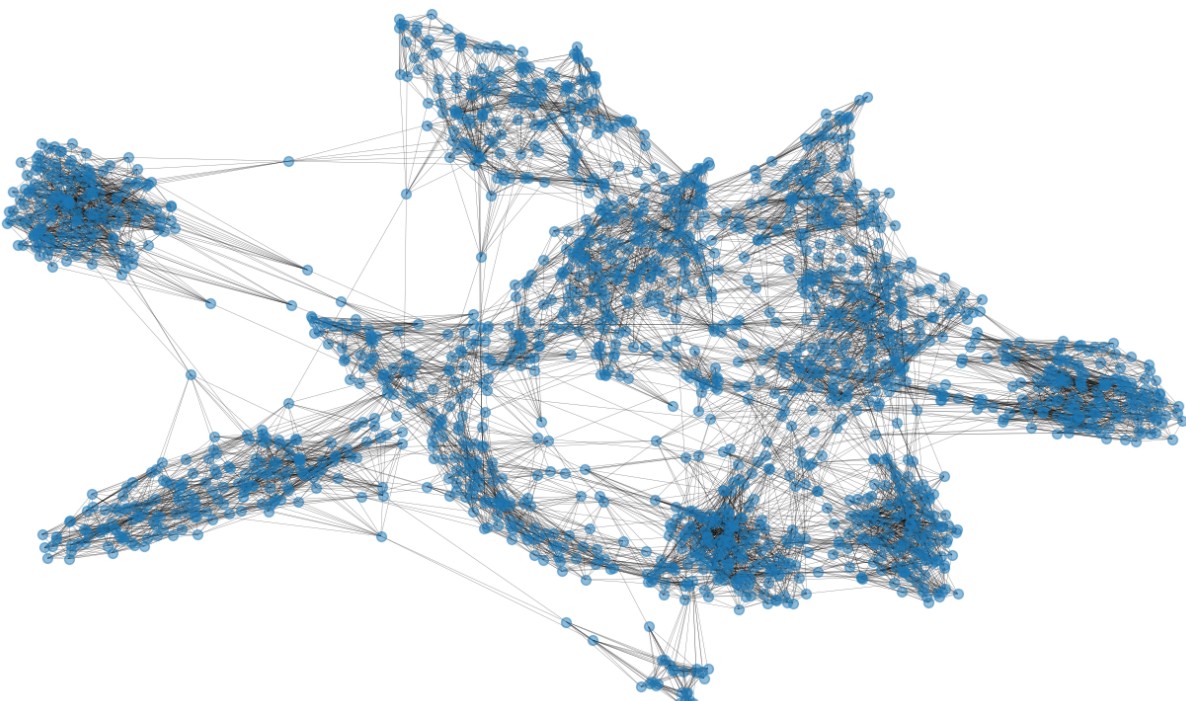

Figure 1: The $k$-NNG of a data set of handwritten digits with $n = 1797$ samples (graph nodes) and $k = \log_2 n$ neighbors.

The $k$-NNG in Figure 1 adopts the Euclidean distance in the computation of the nearest neighbors of each sample $\vec{x}_i$. Let $\eta_i$ be the neighbors of $\vec{x}_i$. Then, a patch $P_i$ may be defined as the set $\{\vec{x}_i \cup \eta_i\}$. It is worth noticing that the cardinality of this set is $k + 1$. In matrix notation, the patch $P_i$ is given by:

$$P_i = [\vec{x}_i, \vec{x}_{i1}, \vec{x}_{i2}, ..., \vec{x}_{ik}] = \begin{bmatrix} x_i(1) & x_{i1}(1) & \ldots & x_{ik}(1) \\ x_i(2) & x_{i1}(2) & \ldots & x_{ik}(2) \\ \vdots & \vdots & \ddots & \vdots \\ \vdots & \vdots & \ldots & \vdots \\ x_i(m) & x_{i1}(m) & \ldots & x_{ik}(m) \end{bmatrix}_{m \times (k+1)} \tag{1}$$

In the proposed method, we approximate the local metric tensor at a point $\vec{x}_i$ as the inverse of the local covariance matrix, $\Sigma_i^{-1}$, estimated from the patch $P_i$. Our motivations for this choice are listed below, following (Li and Tian, 2018; Wang and Sun, 2015; Hair-Jr. et al., 2018):

- **Positive-definiteness:** The covariance matrix is positive-definite, implying that it defines a positive-definite inner product on the space of the data. Similarly, its inverse retains this property, ensuring that it represents a valid metric tensor.

- **Measurement of distances:** The elements of the inverse covariance matrix (i.e., precision matrix) provide information about the distances between points in the feature space. A larger value in the inverse covariance matrix indicates a smaller distance between the corresponding features, while a smaller value indicates a larger distance. This information reflects the relationships and correlations between features in the data set.

- **Directional sensitivity:** Similarly to a metric tensor in differential geometry, the inverse covariance matrix is sensitive to changes in the direction within the feature space. It quantifies how distances between points change as one moves along different directions in the space, capturing the curvature and geometry of the data manifold.

- **Mahalanobis distance:** The inverse covariance matrix is closely related to the Mahalanobis distance, which is a non-isotropic distance between vectors, considering the correlations between features. The role of the inverse covariance matrix in the Mahalanobis distance is to allow different degrees of deformations of the space. It has been applied in various statistical and machine learning problems related to outlier detection, clustering, and classification.

Hence, a discrete approximation for the metric tensor at $\vec{x}_i$ is given by the inverse of the local covariance matrix $\Sigma_i$:

$$\Sigma_i = E[(\vec{x}_j - \vec{x}_i)(\vec{x}_j - \vec{x}_i)^T] = \frac{1}{k} \sum_{x_j \in P_i} (\vec{x}_j - \vec{x}_i)(\vec{x}_j - \vec{x}_i)^T \tag{2}$$

where $\vec{x}_i$ is the central sample. Our approximation consists in setting the first fundamental form at a given point $\vec{x}_i$ as $\mathbb{I}_i \approx \Sigma_i^{-1}$ for every sample in the $k$-NNG.

To compute a discrete approximation for the second fundamental form, we use a mathematical result that states that $\mathbb{II}$ is related to the Hessian matrix through the parametrization of the surface, and the scalar-valued function representing the surface itself. The coefficients of the second fundamental form may be expressed in terms of the second partial derivatives of the components of the surface parametrization function. The second fundamental form is then proportional to the Hessian, which is the matrix of second order partial derivatives (Kuhnel, 2015). Thus, our approach is based in the strategy adopted by the manifold learning algorithm known as Hessian eigenmaps (Donoho and Grimes, 2003). Considering the case of $m = 2$ - meaning that the feature vectors are 2D, the $X_i$ is defined as the matrix composed by the following columns:

$$X_i = \left[1, U_1, U_2, U_1^2, U_2^2, (U_1 \times U_2)\right] \tag{3}$$

where $U_i$ denotes the i-*th* eigenvector of the local covariance matrix, computed from the samples belonging to the patch $P_i$. The notation $U_i \times U_j$ denotes the pointwise product between vectors $U_i$ and $U_j$.

When $m > 2$, the matrix $X_i$ must have $1 + m + m(m+1)/2$ columns, in which the first column is a vector of $1's$, the next $m$ columns are the eigenvectors of $\Sigma_i$, and the final $m(m+1)/2$ columns are the square eigenvectors followed by the several cross products between them. For example, if $m = 3$, the matrix $X_i$ is given by:

$$X_i = \left[1, U_1, U_2, U_3, U_1^2, U_2^2, U_3^2, (U_1 \times U_2), (U_1 \times U_3), (U_2 \times U_3)\right] \tag{4}$$

The main problem is that the columns of $X_i$ are not orthogonal. It is then necessary to orthogonalize them by applying the Gram-Schmidt orthogonalization algorithm (Leon et al., 2013), preparing a new matrix $\tilde{X}_i$. The $H_i$ is then defined as the matrix with the last $m(m+1)/2$ columns of $X_i$ transposed:

$$(H_i)_{r,l} = \left(\tilde{X}_i\right)_{l,1+m+r} \tag{5}$$

A final step consists of transforming the Hessian estimator into a square $m \times m$ matrix. The local second fundamental form at sample $\vec{x}_i$, denoted by $\mathcal{H}_i$, is then given by:

$$\mathcal{H}_i = H_i H_i^T \tag{6}$$

Our approximation becomes $\mathbb{III}_i \approx \mathcal{H}_i$. Both estimators for $\mathbb{I}$ and $\mathbb{III}$ are square $m \times m$ matrices. Lastly, our approximation to the local shape operator at $\vec{x}_i$ is given by:

$$\mathcal{S}_i = -\mathbb{III}_i(\mathbb{I}_i)^{-1} = -\mathcal{H}_i \Sigma_i \tag{7}$$

Therefore, the determinant of $\mathcal{S}_i$ is the desired Gaussian curvature at point $\vec{x}_i$. With the proposed method, it is possible to assign a curvature for each sample of the data set. Algorithm 1 details the pseudocode of the proposed shape operator curvature estimation method.

---

**Algorithm 1** Shape operator based curvatures

---

    **function** SHAPE-OPERATOR-CURVATURES$(X, k)$
        // $X$: the $n \times m$ data matrix (each row is a sample)
        // $k$: the number of neighbors in the $k$NN-graph
        $A \leftarrow k$NN-graph$(X, k)$                               ▷ Builds the $k$NN-graph
        **for** $i \leftarrow 1; i < n; i++$ **do**
            $neighbors \leftarrow N(\vec{x}_i)$                   ▷ Neighborhood of sample $\vec{x}_i$
            $\Sigma_i \leftarrow$ cov-matrix(neighbors)             ▷ Local covariance matrix
            $U \leftarrow eigenvectors(\Sigma_i)$             ▷ Eigenvectors = columns of U
            Compute the matrix $X_i$ with $1 + m + m(m+1)/2$ columns
            $\hat{X}_i \leftarrow$ Gram-Schmidt$(X_i)$          ▷ Gram-Schmidt orthogonalization
            Compute the matrix $H_i$: the last $m(m+1)/2$ columns of $\hat{X}_i$
            $\mathcal{H}_i \leftarrow \hat{H}_i \hat{H}_i^T$                       ▷ Second fundamental form
            $\mathcal{S}_i \leftarrow -\mathcal{H}_i \Sigma_i$                       ▷ Shape operator
            $K_i \leftarrow det(\mathcal{S}_i)$                  ▷ Curvature at point $\vec{x}_i$
        **end for**
        return $K$                                    ▷ Vector of curvatures
    **end function**

---

## 2.2 Curvature-based $kK$-NN classifier

The intuition behind the new $kK$-NN classifier is to explore the local curvature to define the size of the neighborhood ($k$) at each vertex of the $k$-NNG in an adaptive fashion. In the case of points with a lower curvature values, the tangent plane tend to be tightly adjusted to a manifold as the geometry is relatively flat in the neighborhood of that point. This means that a larger parameter $k$ should be considered. Conversely, points with high curvature values lead to a relevant bending or deformation of the manifold. In those cases, it is challenging to approximate accurately with a flat tangent plane. The tangent plane may be loosely adjusted to the surface in such regions, which means that a smaller parameter $k$ should be considered.

The $kK$-NN classifier is composed by two phases, namely training and testing. In the training stage, the first step consists in building the $k$-NNG from the input feature space using $k = log_2\ n$, where $n$ is the number of samples. Then, it is computed the curvature of all vertices of the $k$-NNG exploring the shape operator based

algorithm detailed in the previous section. Subsequently, the curvatures are quantized into 10 different scores - from zero to nine. Given the scores based on the local curvatures, the adaptive neighborhood adjustment is performed by pruning the edges of the $k$-NNG as detailed below.

Suppose that $k = 11$ and sample $\vec{x}_i$ have a curvature score $c_i = 4$. The edges are then disconnected, linking the sample $\vec{x}_i$ and its four farthest neighbors, reducing the neighborhood to only seven neighbors. In the case of $k < c_i$, the sample $\vec{x}_i$ remains connected to its nearest neighbor. In the testing stage, the new sample $\vec{z}_i$ is included to be classified by the $k$-NNG, linking it to its $k$ nearest neighbors and computing its local curvature. Then, the curvature of the new point is added into the curvatures vector, generating its curvature score $c_i$ through a quantization process. Lastly, its farthest $k - c_i$ neighbors are pruned while assigning the label that occurs in the adjusted neighborhood most frequently. The algorithm 2 presents the pseudocode for the training stage of the proposed $kK$-NN classifier.

---

**Algorithm 2** $kK$-NN classifier training

    **function** $kK$NN-TRAIN(train_set, train_labels, $k$)
        curvatures $\leftarrow$ Shape-Operator-Curvatures(train_set, $k$)
        **return** curvatures
    **end function**

---

The computational complexity of the function $kK$NN-train is equivalent to the complexity of the function to compute the curvatures using local shape operators. Subsequently, the algorithm 3 details the pseudocode for the testing stage of the $kK$-NN classifier.

---

**Algorithm 3** $kK$-NN classifier testing

    **function** $kk$NN-TEST(test_samples, train_labels, curvatures, $k$)
        n $\leftarrow$ size(test_samples)
        predictions $\leftarrow$ zeros(n)
        **for** $i \leftarrow 0; i < n; i++$ **do**
            patch $\leftarrow$ Find_Neighbors(test_samples[i], $k$)
            new_curvature $\leftarrow$ Shape-Operator-Curvatures(patch, $k$)
            curvatures $\leftarrow$ concatenate(curvature, new_curvature)
            scores $\leftarrow$ quantize(curvatures, 10)
            new_score $\leftarrow$ scores[-1]                    $\triangleright$ Last score is from the new test sample
            neighborhood $\leftarrow$ Adjust_Neighbors(patch, new_score)
            predictions[i] $\leftarrow$ Majority_Vote(train_labels[neighborhood])
        **end for**
        **return** predictions
    **end function**

---

In the Shape-Operator-Curvatures, the main loop iterates $n$ times, where $n$ is the number of samples in the training set. The complexity of selecting the $k$-nearest neighbors is $O(nmk)$, where $m$ is the dimensionality. The covariance matrix computation has cost $O(nm^2)$, while the regular eigendecomposition methods amounts to a total cost $O(m^3)$. The Gram-Schmidt orthogonalization of a set of $n$ $m$-dimensional vectors is $O(mn^2)$. As the number of columns in $H_i$ is $m(m+1)/2$, which is $O(m^2)$ then the complexity becomes $O(mm^4)$, which results in $O(m^5)$. The regular matrix product between two $m \times m$ matrices has cost $O(m^3)$, and the the computation of the determinant of a $m \times m$ matrix is also $O(m^3)$. Therefore, the total cost for our shape operator based algorithm for curvature computation is:

$$O(n^2mk) + O(n^2m^2) + O(nm^5) \tag{8}$$

illustrating that it may be expressed as $O(nm(nk + nm + m^4))$. In practice, the complexity analysis reveals that the proposed algorithm scales better to an arbitrary increase in the number of samples compared to an arbitrary increase in the number of features. For this reason, for data sets with a large number of features, a dimensionality reduction method such as principal component analysis (PCA) might be required before the computation of the local curvatures.

The computational complexity of the function $kk$NN-test is also dominated by the computation of the curvatures. The function Shape-Operator-Curvatures is responsible for the computation of the the curvature of a simple point. Nonetheless, due to the fact that is inside a FOR loop of size $n$, the global cost is equivalent to the function employed in the training phase. In comparison with the regular $k$-NN, the proposed adaptive $kK$-NN classifier shows a significantly higher computational cost. However, as reported in the subsequent sections, the new $kK$-NN classifier is capable of improving the classification performance in several data sets, particularly when the number of samples in the training set is limited. The fact that the bottleneck of the $kK$-NN classifier is the curvature estimation algorithm, may be considered as a caveat of the proposed method.

## 3 Results

To evaluate the proposed method, computational experiments are performed to compare the balanced accuracy, Kappa coefficient, Jaccard index, and F1-score between the regular $k$-NN and the adaptive $kK$-NN classifier. Several real-world data sets are collected from the public repository `openml.org`. Table 1 reports the names, number of samples, number of features, and number of classes of 30 data sets used in the first round of experiments. It is worth mentioning that those data sets have a wide range of number of samples, features, and classes.

Table 1: Number of samples, features, and classes of the selected openML data sets for the first round of experiments.

| # | Data sets | # samples | # features | # classes |
|---|---|---|---|---|
| 1 | vowel | 990 | 13 | 11 |
| 2 | zoo | 101 | 16 | 7 |
| 3 | thyroid-new | 215 | 5 | 3 |
| 4 | lawsuit | 264 | 4 | 2 |
| 5 | arsenic-male-bladder | 559 | 4 | 2 |
| 6 | tecator | 240 | 124 | 2 |
| 7 | sonar | 208 | 60 | 2 |
| 8 | ionosphere | 351 | 34 | 2 |
| 9 | prnn_crabs | 200 | 7 | 2 |
| 10 | monks-problem-1 | 556 | 6 | 2 |
| 11 | diggle_table_a2 | 310 | 8 | 9 |
| 12 | user-knowledge | 403 | 5 | 5 |
| 13 | tic-tac-toe | 958 | 9 | 2 |
| 14 | parkinsons | 195 | 22 | 2 |
| 15 | glass | 214 | 9 | 6 |
| 16 | breast-tissue | 106 | 9 | 4 |
| 17 | Smartphone-Based_Recognition | 180 | 66 | 6 |
| 18 | FL2000 | 67 | 15 | 5 |
| 19 | fishcatch | 158 | 7 | 2 |
| 20 | biomed | 209 | 8 | 2 |
| 21 | kidney | 76 | 6 | 2 |
| 22 | anneal | 898 | 38 | 5 |
| 23 | mfeat-fourier (25%) | 500 | 76 | 10 |
| 24 | mfeat-karhunen (25%) | 500 | 64 | 10 |
| 25 | letter (10%) | 2000 | 16 | 26 |
| 26 | satimage (25%) | 1607 | 36 | 6 |
| 27 | pendigits (25%) | 2748 | 16 | 10 |
| 28 | texture (25%) | 1375 | 40 | 11 |
| 29 | digits (25%) | 449 | 64 | 10 |
| 30 | Olivetti_Faces (10 LDA features) | 400 | 10 | 40 |

In the first round of experiments, the methodology adopts a holdout strategy to divide the samples into the training and test data sets. The training partition varies from 10% of the total samples to 90% of the total samples, using increments of 5% - leading to a total of 17 possible divisions in training and testing. Both the regular $k$-NN and the proposed $kK$-NN are trained in each one of the 17 training sets, while testing them in the respective test partition. The median balanced score, Kappa coefficient, Jaccard index, and F1-score are computed over the 17 executions. The objective is to test the behavior of the classifier in all possible scenarios - small, medium, and large training data sets. The results are reported in Table 2. The proposed $kK$-NN classifier outperforms the regular $k$-NN by a significant margin, considering all the evaluation metrics.

Table 2: Median of measures after 17 executions adopting the holdout strategy with training data sets of different sizes: from 10% to 90% of the number of samples with increments of 5%

| | $k$-NN | | | | $kK$-NN | | | |
|---|---|---|---|---|---|---|---|---|
| # | Bal. Acc. | Kappa | Jacc. | F1 | Bal. Acc. | Kappa | Jacc. | F1 |
| 1 | 0.5314 | 0.4679 | 0.343 | 0.4978 | **0.8860** | **0.8732** | **0.7999** | **0.8828** |
| 2 | 0.5622 | 0.6898 | 0.6592 | 0.7136 | **0.8809** | **0.9113** | **0.8924** | **0.9337** |
| 3 | 0.7525 | 0.7077 | 0.8039 | 0.8840 | **0.8746** | **0.8628** | **0.8927** | **0.9417** |
| 4 | 0.5555 | 0.1889 | 0.8701 | 0.9070 | **0.8758** | **0.7517** | **0.9382** | **0.9662** |
| 5 | 0.5000 | 0.0000 | 0.9137 | 0.9354 | **0.7884** | **0.6095** | **0.9354** | **0.9599** |
| 6 | 0.7779 | 0.5698 | 0.6479 | 0.7843 | **0.8308** | **0.6689** | **0.7142** | **0.8333** |
| 7 | 0.7227 | 0.4547 | 0.5712 | 0.7240 | **0.8285** | **0.6599** | **0.7121** | **0.8315** |
| 8 | 0.7326 | 0.5146 | 0.6568 | 0.7839 | **0.8205** | **0.6824** | **0.7484** | **0.8534** |
| 9 | 0.8097 | 0.6196 | 0.6806 | 0.8099 | **0.9677** | **0.9334** | **0.9354** | **0.9666** |
| 10 | 0.7083 | 0.4112 | 0.5438 | 0.7044 | **0.8140** | **0.6188** | **0.6748** | **0.8055** |
| 11 | 0.8446 | 0.8316 | 0.7576 | 0.8530 | **0.9376** | **0.9341** | **0.9047** | **0.9436** |
| 12 | 0.5371 | 0.6021 | 0.5498 | 0.6791 | **0.5872** | **0.6661** | **0.6085** | **0.7405** |
| 13 | 0.6619 | 0.3828 | 0.5927 | 0.7261 | **0.7146** | **0.4468** | **0.6398** | **0.7710** |
| 14 | 0.7324 | 0.5573 | 0.7558 | 0.8510 | **0.9441** | **0.8336** | **0.8938** | **0.9430** |
| 15 | 0.3938 | 0.4112 | 0.4238 | 0.5692 | **0.5429** | **0.4746** | **0.4581** | **0.6194** |
| 16 | 0.3731 | 0.1666 | 0.3175 | 0.4567 | **0.4802** | **0.2557** | **0.3631** | **0.5227** |
| 17 | 0.8720 | 0.8327 | 0.7712 | 0.8635 | **0.9126** | **0.8971** | **0.8503** | **0.9161** |
| 18 | 0.3144 | 0.2000 | 0.3948 | 0.5270 | **0.4251** | **0.4819** | **0.5647** | **0.7005** |
| 19 | 0.9555 | 0.9141 | 0.9192 | 0.9578 | **0.9813** | **0.9626** | **0.9646** | **0.9819** |
| 20 | 0.8472 | 0.7448 | 0.8036 | 0.8884 | **0.8750** | **0.7976** | **0.8375** | **0.9104** |
| 21 | 0.6589 | 0.3132 | 0.4859 | 0.6536 | **0.7652** | **0.4976** | **0.5793** | **0.7333** |
| 22 | 0.5703 | 0.7461 | 0.8307 | 0.8945 | **0.7719** | **0.7996** | **0.8577** | **0.9187** |
| 23 | 0.6725 | 0.6481 | 0.5596 | 0.6890 | **0.6837** | **0.6573** | **0.5809** | **0.6961** |
| 24 | 0.8566 | 0.8494 | 0.7700 | 0.8649 | **0.8640** | **0.8595** | **0.7817** | **0.8735** |
| 25 | 0.6210 | 0.6002 | 0.456 | 0.6120 | **0.6901** | **0.6740** | **0.5359** | **0.6873** |
| 26 | 0.7977 | 0.7954 | 0.7303 | 0.8309 | **0.8405** | **0.8236** | **0.7638** | **0.8557** |
| 27 | 0.9579 | 0.9546 | 0.9221 | 0.9589 | **0.9830** | **0.9821** | **0.9687** | **0.9839** |
| 28 | 0.9123 | 0.9087 | 0.8525 | 0.9165 | **0.9429** | **0.9412** | **0.9023** | **0.9468** |
| 29 | 0.8851 | 0.8731 | 0.8097 | 0.8855 | **0.9261** | **0.9175** | **0.8686** | **0.9251** |
| 30 | 0.7821 | 0.7029 | 0.5857 | 0.6402 | **0.9949** | **0.9935** | **0.9885** | **0.9936** |
| Mean | 0.6966 | 0.5886 | 0.6660 | 0.7687 | **0.8143** | **0.7489** | **0.7719** | **0.8546** |
| Median | 0.7276 | 0.6109 | 0.6699 | 0.7971 | **0.8523** | **0.7986** | **0.8187** | **0.8966** |
| Smallest | 0.3144 | 0.0000 | 0.3175 | 0.4567 | **0.4251** | **0.2557** | **0.3631** | **0.5227** |
| Largest | 0.9579 | 0.9546 | 0.9221 | 0.9589 | **0.9949** | **0.9935** | **0.9885** | **0.9936** |

Figure 2 illustrates the curves of the balanced accuracies obtained for the data sets `vowel` and `Olivetti_Faces`. This is to visualize how the proposed method performs in a single data set. The $kK$-NN classifier is consistently superior compared to the regular $k$-NN. The rationale behind the capacity of

the proposed method to improve the regular $k$-NN may be summarized by three relevant aspects. Firstly, through the use of an adaptive strategy to define the neighborhood sizes, the $kK$-NN is more successful in avoiding both underfitting and overfitting. Secondly, in more dense regions the decision boundaries become more adjusted to the samples, while and in less dense regions the boundaries become smoother - making the classification rule adaptive to different regions of the feature space. Thirdly, high curvature points are commonly related to outliers, being the $kK$-NN classifier capable of isolating such samples by drastically reducing its neighborhoods - which, consequently, decreases its overall sensitivity to outliers.

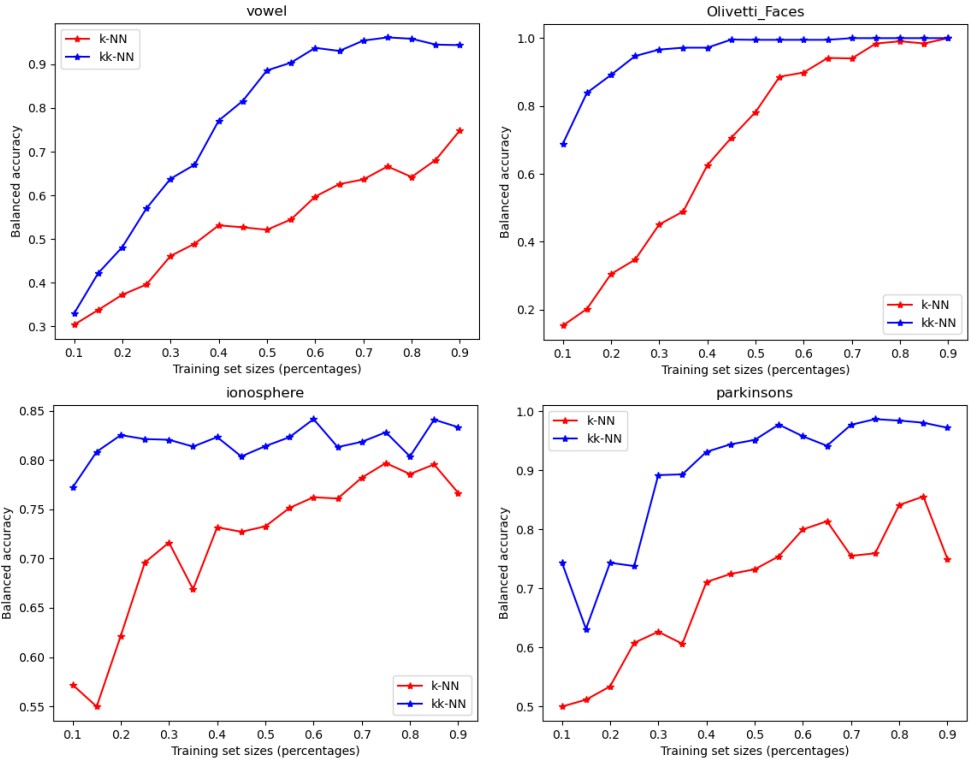

Figure 2: The balanced accuracy curves built by the holdout strategy for the regular $k$-NN and proposed $kK$-NN classifiers using different partition sizes, from 10% to 90% of the total number of samples. Top-left: `vowel` data set. Top-right: `Olivetti_Faces` data set. Bottom-left: `ionosphere` data set. Bottom-right: `parkinsons` data set.

In the second round of experiments, the objective is to compare the behavior of the regular $k$-NN and the proposed $kK$-NN when the training set is reduced. The results indicate an interesting feature of the adaptive curvature based classifier, namely its capacity of learning from a limited number of samples. Table 3 reports the selected data sets and their number of samples, features, and classes.

Table 3: Number of samples, features, and classes of the selected openML data sets for the second round of experiments.

| # | Data sets | # samples | # features | # classes |
|---|---|---|---|---|
| 1 | UMIST_Faces_Cropped | 575 | 10304 | 20 |
| 2 | variousCancers_final | 383 | 54675 | 9 |
| 3 | micro-mass | 360 | 1300 | 10 |
| 4 | collins | 500 | 22 | 2 |

In the first three data sets in Table 3, the number of features is much larger than the number of samples. Due to the computational complexity of the proposed curvature estimation method, the direct application of the $kK$-NN classifier is not an option for the raw data. To reduce the number of features, a Linear Discriminant Analysis (LDA) is then applied to extract the maximum possible number of features - i.e., the number of classes minus one.

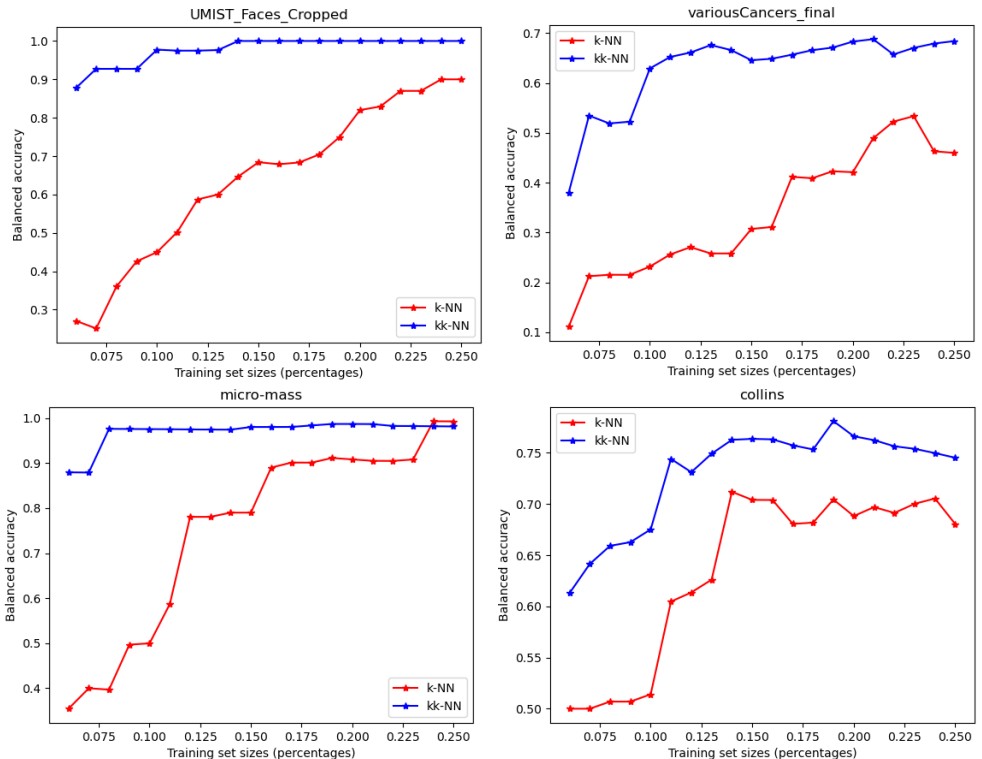

Figure 3: The balanced accuracy curves built by the holdout strategy for the regular $k$-NN and the proposed $kK$-NN classifiers using different partition sizes, from 5% to no more than 40% of the total number of samples. Top-left: `UMIST_Faces_Cropped` data set. Top-right: `variousCancers_final` data set. Bottom-left: `micro-mass` data set. Bottom-right: `collins` data set.

The same strategy as before is performed, but in this round focusing on training sets with no more than 25% of the total number of samples. Figure 3 shows the results for these four data sets. The proposed $kK$-NN classifier is capable of learning more discriminant decision functions when the number of training samples is significantly reduced, indicating that the proposed method may do more with less. These results indicate that the $kK$-NN algorithm is able to produce more flexible and adjustable decision boundaries while reducing the influence of outliers over the classification process.

## 4   Conclusion

In the present work, we introduce a new curvature-based classification method. The $kK$-NN consists of an adaptive approach to overcome relevant limitations of the widely adopted $k$-NN classification algorithm. The rationale behind the new $kK$-NN is to adapt the neighborhood size locally, leveraging the curvature information inherent in the data set to improve classification accuracy. Our theoretical analysis and experimental results provide relevant insights into the effectiveness and versatility of the proposed method. Our main findings over 30 real-world data sets may be summarized into three main methodological improvements.

The first methodological improvement concerns a curvature-based adaptation, in which the embodiment of curvature information into the $k$-NN framework may improve classification performance. By dynamically

adjusting the neighborhood size based on local curvature characteristics, the $kK$-NN exhibits adaptability to different regions of the feature space. The second methodological improvement refers to robustness and generalization. The $kK$-NN classifier exhibits robustness to noise and outliers, indicating its ability to handle complex and real-world data sets more effectively. Moreover, the proposed approach showcases promising generalization capabilities across different domains, emphasizing its potential for a wide range of applications. Thirdly, extensive experiments exploring diverse data sets have shown that the $kK$-NN outperforms established $k$-NN classifiers in various scenarios. The adaptive nature of the $kK$-NN allows it to excel in situations where the local density of samples and curvature patterns vary significantly.

Thus, the adaptive curvature-based approach adopted in the $kK$-NN introduces a promising advancement within $k$-NN classification methods. Empirical evidence supports its efficacy in diverse scenarios, indicating relevant avenues for future research and applications in machine learning, pattern recognition, computer vision, among other domains. The incorporation of curvature information into classification frameworks offers a nuanced perspective that unfolds new possibilities for enhancing the adaptability and robustness of classification algorithms.

Suggestion of future extensions include further theoretical investigations into the properties of curvature-adaptive classifiers that should provide a deeper understanding of its underlying mechanisms. Analyzing the convergence properties and establishing theoretical bounds on the performance of $kK$-NN under different conditions would contribute to the theoretical foundations of curvature-based classification. In addition, the curvature-based adaptation of the $kK$-NN could be applied to image processing tasks, particularly in scenarios where local variations and intricate patterns play a central role. Moreover, as the principles underlying $kK$-NN are rooted in curvature analysis, the new method may find applications in dimensionality reduction and metric learning.

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

# *Appendix*

## A  Shape operator and curvatures

Differential geometry provides a framework to study properties that are invariant under smooth mappings, preserving differentiability. It focuses on studying geometric objects such as curves and surfaces, while understanding their properties regarding differential calculus (do Carmo, 2017; Needham, 2021). Potential applications across various domains include physics, engineering, computer science, robotics, among many more. In physics, it is applied to describe the geometry of spacetime in general relativity, whereas in computer graphics it is employed to model and manipulate smooth surfaces (Tu, 2017). The mathematical concepts and tools of differential geometry provide a powerful framework to understand the intrinsic geometry of spaces as well as their applications in diverse fields of knowledge (Oprea, 2007). The notion of surface in $\mathbb{R}^3$ is generalized to higher dimensions by the definition of manifold.

**Definition 1** (Manifold)**.** *Let $M$ be a topological space. Then, $M$ is an n-dimensional smooth manifold if it satisfies the following conditions:*

- ***Locally Euclidean:*** *for every point $p \in M$, there exists an open neighborhood $U$ of $p$ such that $U$ is homeomorphic to an open subset of $\mathbb{R}^n$. Formally, there exists a homeomorphism $\phi : U \to V$, where $V$ is an open subset of $\mathbb{R}^n$.*

- ***Smoothness:*** *The collection of such local homeomorphisms $\phi$ forms an atlas on the manifold $M$. A smooth manifold $M$ is equipped with a smooth structure, which consists of an atlas whose transition maps (maps between overlapping neighborhoods) are all smooth, meaning that they have continuous derivatives of all orders.*

- ***Hausdorff and second-countable:*** *The manifold $M$ is required to be a Hausdorff space, meaning that for any two distinct points in $M$, there exist disjoint open sets containing each point. Moreover, $M$ must be second-countable, which means that its topology has a countable basis.*

Manifolds serve as a fundamental framework for studying spaces with intrinsic geometry, finding applications in many areas of mathematics, including data analysis and machine learning, where they are employed to model complex data sets and high-dimensional spaces (Gorban and Tyukin, 2018; Fefferman et al., 2016).

A tangent space is a fundamental concept in differential geometry, providing an understanding of the local behavior of a manifold at a particular point. The tangent space to a manifold at a point captures the notion of "infinitesimal" directions at that point. Tangent vectors represent directions and rates of change at a point on the manifold. Intuitively, in a smooth surface such as a sphere or curve, the tangent vectors at a particular point represent possible directions to move along the surface or the direction of velocity if it is passing through that point.

**Definition 2** (Tangent space)**.** *Let $M$ be a smooth manifold and $p$ a point in $M$. The tangent space $T_pM$ to $M$ at $p$ is defined as the set of all tangent vectors at $p$.*

- ***Basis of tangent space:*** *The tangent space $T_pM$ is a vector space, which basis consisting of tangent vectors corresponding to coordinate curves passing through $p$. In local coordinates, these basis vectors are frequently denoted as $\partial/\partial x^i$, where $i$ ranges over the $n$ dimensions of the manifold $M$.*

The tangent space captures the local geometry of the manifold at a specific point, being instrumental in defining notions such as tangent bundles, differential forms, and differential operators on manifolds. Another mathematical object that plays an important role in the computation of inner products and arc lengths in a manifold is the metric tensor, also known as the first fundamental form.

**Definition 3** (First fundamental form or metric tensor). *Let $M$ be a smooth manifold. The first fundamental form on $M$, denoted by $\mathbb{I}$ or $g$, is a smoothly varying collections of inner products defined on the tangent spaces of $M$, such that for each point $p \in M$ the metric tensor defines an inner product $g_p$ to the tangent space $T_pM$. The metric tensor $g$ is a symmetric, non-degenerate, and smoothly varying bilinear form on the tangent bundle of the manifold. This means that for any two tangent vectors $X, Y$ at a point $p$ on $M$, the metric tensor $g$ assigns a real number $g_p(X, Y)$ that satisfies:*

- ***Symmetry:*** *$g_p(X, Y) = g_p(Y, X)$ for all the tangent vectors at $p$.*

- ***Linearity:*** *$g_p$ is linear in each argument, thus for any tangent vectors $X, Y Z$ and scalars $\alpha, \beta$, we have:*

$$g_p(\alpha X + \beta Y, Z) = \alpha g_p(X, Z) + \beta g_p(Y, Z) \tag{9}$$
$$g_p(X, \alpha Y + \beta Z) = \alpha g_p(X, Y) + \beta g_p(X, Z) \tag{10}$$

- ***Non-degeneracy:*** *The metric tensor is non-degenerate, meaning that for any non-zero tangent vector $X$ at $p$, there exists another tangent vector $Y$ such that $g_p(X, Y) \neq 0$.*

Intuitively, the first fundamental form enables the computation of distances along the paths within the surface. It is known as an intrinsic metric due to the fact that is a Riemannian metric of the manifold. The second fundamental form of a manifold is a geometric object that characterizes the extrinsic curvature of a submanifold within a higher-dimensional manifold. This is also known as the Euler-Schouten embedding curvature. Commonly, the submanifold $M$ with $m$ dimensions is embedded in an Euclidean space $\mathbb{R}^n$, with $n > m$ (see Nash embedding theorems).

The second fundamental form is a pivotal instrument in the study of submanifolds (Chen, 2019), providing important geometric information about their shape and curvature within the ambient manifold. It plays a significant role in various areas of mathematics, including differential geometry, geometric analysis, mathematical physics, and computer vision (Yilmaz and Shah, 2005).

**Definition 4** (Second fundamental form). *Let $M$ be a smooth manifold. The second fundamental form of $M$, denoted by $\mathbb{III}$, is a bilinear form defined on the tangent space of $M$ at each point. Basically, it is related to the curvature of the normal section along a direction $\vec{v}$ at the point $p$. In simpler terms, it measures how curved the trajectory should be if it is moving along the direction $\vec{v}$. The second fundamental form reveals how fast the manifold moves away from the tangent plane).*

The second fundamental form describes how curved the embedding is, indicating how the manifold is located within the ambient space. It is a type of derivative of the unit normal along the surface. Equivalently, it is the rate of change of the tangent planes taken in various directions, consisting of an extrinsic quantity.

**Definition 5** (Shape operator and curvatures). *The shape operator of a manifold $M$ with first fundamental form $\mathbb{I}$ and second fundamental for $\mathbb{III}$ is given by $P = -\mathbb{III} \cdot \mathbb{I}^{-1}$.*

- *The Gaussian curvature $K_G$ is the determinant of the shape operator $P$.*

- *The mean curvature $K_M$ is the trace of the shape operator $P$.*

- *The principal curvatures are the eigenvalues of the shape operator $P$.*

Considering the definitions above, we introduce an algorithm for curvature estimation through a nonparametric approximation for the shape operator. This new algorithm adopts local curvatures to adjust the neighborhoods within the $k$-NN classifier through an adaptive approach.

# B   The $k$-nearest neighbor classifier

The $k$-nearest neighbor ($k$-NN) classifier is a simple yet effective algorithm employed in supervised learning. It belongs to the family of instance-based learning or lazy learning methods, where the algorithm does not build an explicit model during the training phase. Instead, it memorizes the entire training data set to perform predictions based on the similarity of new instances compared to the training data set (Taunk et al., 2019).

The $k$-NN may be understood as the Bayesian classifier when the conditional class densities are estimated using a nonparametric approach (Webb and Copsey, 2011). From probability theory, we know that the probability of a sample $\vec{x}$ be in a region of interest of volume $V(\vec{x})$, centered on $\vec{x}$ is given by:

$$\theta = \int_{V(\vec{x})} p(\vec{x})d\vec{x} \tag{11}$$

which is a generalization of the area under the curve defined by the probability density function (PDF). In the case of multiple dimensions, a volume is calculated instead of an area. However, for a small volume $V(\vec{x})$, we have the following approximation:

$$\theta \approx p(\vec{x})V(\vec{x}) \tag{12}$$

as the the volume of the box may be computed as the volume of the region times the height (i.e., probability). The probability $\theta$ may be approximated by the proportion of samples that belong to the region of volume $V(\vec{x})$. In the case there are $k$ samples from a total of $n$ that belong to such a region, then:

$$\theta \approx \frac{k}{n} \tag{13}$$

which is a proportion and therefore belongs to the interval $[0, 1]$. Combining equations 12 and 13, we have an approximation for $p(\vec{x})$:

$$p(\vec{x}) = \frac{k}{nV(\vec{x})} \tag{14}$$

Hence, we may estimate the conditional probability of the class $\omega_j$ in a nonparametric manner as follows:

$$p(\vec{x}|\omega_j) = \frac{k_j}{n_j V_R} \tag{15}$$

where $k_j$ denotes the number of samples in class $\omega_j$ in the region of interest, $n_j$ denotes the total number of samples in class $\omega_j$, and $V_R$ represents the volume of the region of interest. Similarly, the prior probability of class $\omega_j$ is given by:

$$p(\omega_j) = \frac{n_j}{n} \tag{16}$$

where $n$ denotes the total number of samples. Therefore, by the maximum a posteriori criterion (MAP), we must assign the sample $\vec{x}$ to the class $\omega_j$ if:

$$p(\omega_j|\vec{x}) > p(\omega_i|\vec{x}) \qquad \forall i \neq j \tag{17}$$

which leads to:

$$\frac{k_j}{n_j V_R} \frac{n_j}{n} > \frac{k_i}{n_i V_R} \frac{n_i}{n} \qquad \forall i \neq j \tag{18}$$

and after some simplifications, it is reduced to:

$$k_j > k_i \qquad \forall i \neq j \tag{19}$$

This means that the sample $\vec{x}$ should be assigned to the most representative class in its $k$ nearest neighbors. Algorithms 4 and 5 detail the pseudocodes of the regular $k$-NN classifier.

---

**Algorithm 4** Get the $k$ nearest neighbors of a given sample

---
   **function** GET_NEIGHBORS(train_set, test_row, k)
      distances ← [ ]
      indices ← [ ]
      **for** each train_row in train_set **do**
         dist ← EuclDist(test_row, train_row[:-1])         ▷ Last feature is the class label
         distances.append(train_row)
      **end for**
      Sort the tuples in distances in ascending order of dist
      neighbors ← [ ]
      **for** $(i = 0; i < k; i{++})$ **do**
         neighbors.append(distances[i][0])
      **end for**
      **return** neighbors
   **end function**

---

**Algorithm 5** $k$-NN classification

---
   **function** KNN_CLASSIFICATION(train_set, test_row, k)
      neighbors ← get_neighbors(train_set, test_row, k)
      labels ← [row[-1]] for row in neighbors]         ▷ Last feature is the label
      prediction ← mode(labels)         ▷ The label that occurs more often
      **return** prediction
   **end function**

---

It has been reported that the probability of error in the $k$-NN classifier is directly related to the probability of error in the Bayesian classifier - which is the lowest among supervised classifiers (Cover and Hart, 1967):

$$P_{knn} \leq P^* \left( 2 - \frac{c}{c-1} P^* \right) \tag{20}$$

where $c$ denotes the number of classes and $P^*$ is the probability of error in the Bayesian classifier. The computational complexity of an $k$-NN prediction according to Algorithm 4 is $O(mn \log n + mk)$, where $n$ is the number of samples, $m$ is the number of features, and $k$ is the number of neighbors. Due to the presence of a sorting procedure, it is possible to perform an $k$-NN prediction in $O(n(m + k))$.

It is worth noticing that there is an intrinsic relation between the $k$-NN classifier and Voronoi tessellation. This relation lies in their geometric interpretation as well as adoption of partitioning the feature space

(Mérigot et al., 2010). Figure 4, originally from (Fortmann-Roe, 2012), depicts an example of a Voronoi diagram in the partition of an 2D feature space.

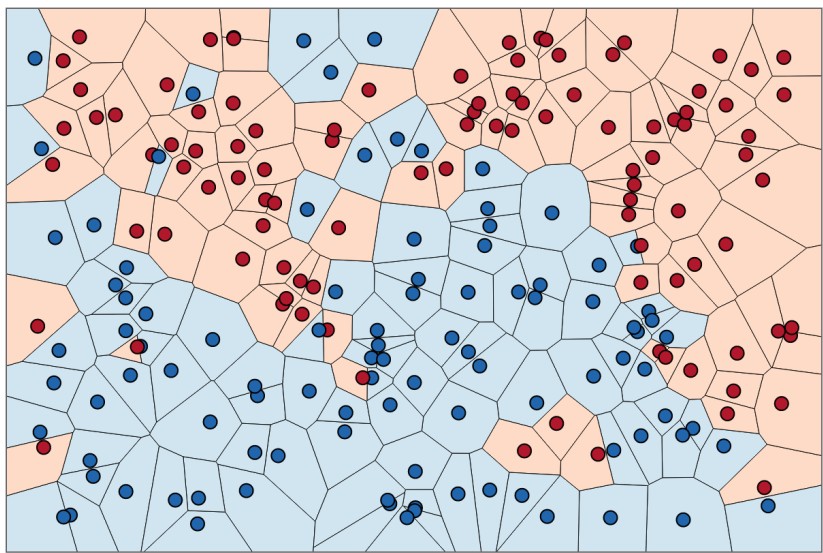

Figure 4: The $k$-NN decision boundary and the Voronoi tesselation in an 2D feature space (Fortmann-Roe, 2012)

The decision boundary obtained by the $k$-NN classifier is the union of several piece-wise linear edges of the Voronoi tesselation, which frequently leads to a complex nonlinear behavior.

## B.1 Balloon and sample-point estimator

Adopting only one bandwidth may provide suboptimal results when considering the whole domain. This approach may induce to oversmoothing in high density neighbourhoods. Conversely, in the case of small sample sizes, it may lead to undersmoothing in the neighbourhood of the extreme regions of the distribution (tails). Without a relevant understanding of the density function, it is challenging to select the ideal bandwidth. To mitigate this issue, locally adaptive approaches empower the bandwidth to vary over the domain of the PDF (Jarosz, 2008). One of such locally adaptive approaches refers to the balloon estimator, which was originally proposed as a $k$-NN estimator (Loftsgaarden and Quesenberry, 1965). A general form of the balloon estimator is detailed as follows:

$$\hat{p}(\vec{x}) = \frac{1}{Nh(\vec{x})} \sum_{i=0}^{N-1} K\left[\frac{\vec{x} - \vec{x_i}}{h(\vec{x})}\right] \tag{21}$$

where $N$ is the number of samples, $h(\vec{x})$ is the bandwidth as a function of $\vec{x}$, and $K$ refers to its kernel. Nonetheless, the balloon estimator is subject to several inefficiencies, particularly regarding univariate data. Whether applied globally, its estimate commonly does not integrate to one over the domain. An additional issue refers to the fact that the bandwidth consists of a discontinuous function, which affects the associated PDF (Terrell and Scott, 1992).

A further local bandwidth estimator is the sample-point estimator, which general form is detailed as follows:

$$\hat{p}(\vec{x}) = \frac{1}{N} \sum_{i=0}^{N-1} \frac{1}{h(\vec{x_i})} K\left[\frac{\vec{x} - \vec{x_i}}{h(\vec{x_i})}\right] \tag{22}$$

The difference between the balloon and sample-point estimator lies in the bandwidth $h(\vec{x_i})$, which is a function of $\vec{x_i}$ instead of $\vec{x}$. In the sample-point estimator approach, every data point is assigned a kernel. However, the size of such kernels may change across data points. The sample-point estimator exhibits advantages compared to the balloon estimator. Considering that each kernel is normalized, the estimator consists of an PDF that integrates to one. Moreover, the sample-point estimator may be entirely continuous due to the fact that it adopts the differential attributes of the respective kernel functions (Jarosz, 2008).

