# OpenReview forum: "Adaptive $k$-nearest neighbor classifier based on the local estimation of the shape operator"
_TMLR — Rejected by TMLR_

### Review · Reviewer_GhjB · 2024-05-17

**Summary Of Contributions:**

In this article, the authors propose an improvement of the famous $k$-nearest neighbour classifier with an adaptive $k$. The idea is that when we want to classify a new sample, we should first look at the local curvature of the training data around this new sample and choose the neighbourhood size $k$ accordingly. If the curvature is high, we will choose a small $k$; if the curvature is low, we will choose a larger $k$. Considering smaller neighbourhoods when the curvature is high should avoid underfitting.

In the classical $k$-nearest neighbour algorithm, the choice of $k$ is crucial, so there have been many attempts to make this parameter adaptive. As far as I know, this is the first work on adaptive neighbourhoods based on curvature analysis.

To achieve this, the first step is to estimate the local curvature. The authors suggest looking at the Gaussian curvature $K$. To do this, they suggest estimating the first ($\mathrm{I}$) and second ($\mathrm{II}$) fundamental forms at a given data point and taking $\hat{K}=\det(-\hat{\mathrm{II}}(\hat{\mathrm{I}})^{-1})$.

To estimate the first fundamental form at a given point, they use the empirical precision matrix of the $k$ nearest neighbours with $k=\log_2(n)$. To estimate the second fundamental form, they use an estimate of the Hessian matrix of the surface parameterisation. Both fundamental form estimators seem to have been known for at least a few years.

The training step consists of calculating the curvatures for each training point. To classify a new sample, we first estimate the curvatures at that point with $k=\log_2(n)$, then we quantize all the curvatures into 10 different scores. This score is then used to choose the number of nearest neighbours that will be used in the majority vote that classifies the new point.

The theoretical part concludes with the calculation of the cost of this adaptive algorithm.

The last part is dedicated to an experimental study of the performance of this adaptive algorithm. The experiment is carried out on 30 different data sets, and the adaptive algorithm is compared with the classical $k$-nearest neighbour algorithm using four different metrics (balanced accuracy, Kappa coefficient, Jaccard index, and F1 score). The experiment is carried out for 17 different training set sizes. In light of these results, the adaptive version seems to be much more efficient than the classical one, especially when the training set is small.

**Audience:**

Yes

**Broader Impact Concerns:**

No concern

**Claims And Evidence:**

Yes

**Requested Changes:**

The main changes to be made are:

1) Explaining and justifying the quantization of the Gaussian curvature in 10 different scores, in particular explaining how the quantization is done.

2) Improve the experimental part by comparing the performance of the proposed adaptive algorithm with other adaptive $k$-nearest neighbour classifiers.

3) The code should be publicly available for better reproducibility.

Other formal changes that I think would improve the paper:

* Abstract : The first sentence is useless

* Introduction :
  * The first three paragraphs about parametric/non-parametric don't really belong in this article, they could be shortened or removed.
  * The patch $P_i$ is mentioned in the introduction but defined much later, it's not a good idea to use a notation before defining it.
  * I think you should mention the contents of your appendices at the end of the introduction.
* Subsection 2.1
  * It would be clearer to define $\eta_i$ as the *k nearest* neighbours of $x_i$.
  * Equation (2) I would remove $E(x_j-x_i)(x_j-x_i)^T$ as $\Sigma_i$ is not the real covariance matrix but an empirical covariance matrix.
  * Equations (3) and (4), the notation $X_i$ is confusing with the notation $X$ for the data set, one of the two notations should be changed.
 * Subsection 2.2
   * Last sentence on page 6 "Then, it is computed the curvature of all vertices of the k-NNG exploring the shape operator based algorithm..." seems strange to me, maybe you could write "Then, the curvature of all vertices of the k NNG *is computed using* the shape operator based algorithm...".
    * First paragraph on page 7 "The edges are then disconnected, linking the sample $x_i$ and its four farthest neighbors, reducing the neighborhood to only seven neighbors." should probably read "The edges linking the sample $x_i$ and its four farthest neighbors are then disconnected, reducing the neighborhood to only seven neighbors".

* Appendix A : "It is known as an intrinsic metric due to the fact that is a Riemannian metric of the manifold." should be " It is known as an intrinsic metric due to the fact that *it* is a Riemannian metric of the manifold."

* Appendix B :
   * Perhaps the computational complexity of the k-NN should be in the main part of the article to compare with the computational complexity of the kK-NN.
  * The last section seems to be written in a different style, the term "bandwidth" is used only in this part, which is confusing. We also have no details about the kernel $K$.

**Strengths And Weaknesses:**

**Strengths :**

* The article is written in a very pedagogical way. In the first appendix they propose a brief overview of the important differential geometry notions used in this article, thanks to this appendix the article is self-contained.

* They propose a simple but clever idea to make the classical $k$-nearest neighbour adaptable. They state clearly how they proceed to estimate the Gaussian curvature and clearly explain their choice of estimator.

* The experiments are carried out on a large number of datasets offering a wide range of number of features, samples and classes. They also tried a wide range of training set sizes. Overall, the comparison between k-NN and kK-NN is done with a lot of precision.

**Weaknesses :**

* The article lacks theoretical results, the performance of the proposed methods is only evaluated by experiments. However, these experiments only provide a comparison between the classical (non-adaptive) $k$-nearest neighbour classifier and the proposed adaptive version. A similar comparison with at least one other adaptive approach, as the ones described in Appendix B, could greatly improve the article.

* The quantization of curvatures into 10 different scores is not justified. Why 10 and not 20? How will the choice of the scale of the scores affect the performance of the algorithm? We don't even know if the quantisation is uniform or not.

---

### Review · Reviewer_DNsn · 2024-05-25

**Summary Of Contributions:**

The develops a method for k-NN classification where the value of k is adapted locally.  The method is inspired by the curvature induced by the k-NN graph.

**Audience:**

Yes

**Claims And Evidence:**

No

**Requested Changes:**

Weaknesses 1-5 would need to be satisfactorily addressed.

**Strengths And Weaknesses:**

Strengths:
 1. The problem of locally adapting k for k-NN classifiers is a useful problem to study.

 2. I like that the proposed solution is ultimately fairly simple.


Weaknesses
 1. The experimental section did not explain which values of k were used.  The closest I could find was a "Suppose that k = 11 ... " on page 7.  Were these chosen differently for each dataset, or what this the same across all data sets?  Was the k for k-NN and kK-NN always the same?

 2.  If kK-NN always uses the same initial k as k-NN, then it seems that the one ultimately used for the classification task in kK-NN is smaller for each local neighborhood than for k-NN.  Can the reported improvement be realized by simply using a smaller k' < k in the (k')-NN algorithm?

 3.  Related to 2, there are no experiments that show how this varies with the choice of k.  For a paper on this topic, this should be required, or at least explained in detail why this was not considered.

 4.  How does the quantization scheme work?  This seems a crucial step, but is not explained.

 5.  Algorithm 5 seems to depend on the order of the data points.  If I see a lot of large curvature early, it may affect the production of a point seem afterwards.  But if I see a bunch of small curvature points early, it may give a different result?  Is this indeed the case?  IF so, did you experiment with changing the order of the points?  Could there be a version with similar effectiveness that does not depend on the ordering of the points?

 6.  The estimate of the curvature in high dimension (most data sets considered have more than 10 features), is not easy, and probably requires a lot of samples in any local neighborhood to get a reliable estimate (e.g., $\Omega(m^2)$ in $m$ dimensions, and probably with a decent coefficient).  So if one wanted to estimate curvature, probable we would need k > m^2, which is ofter approaching the size of the data sets used.  How can this be reconciled with the values of k used in the paper?

---

### Review · Reviewer_eVPB · 2024-05-31

**Summary Of Contributions:**

This paper suggests a new method for computing the parameter k adaptively for k-NN. The key idea is to define and compute a notion of curvature for each data point, which reflects the local geometry of them (and this is for the training process). For the testing process, similar curvature is also computed for the testing dataset. The eventual k-NN is performed using calibrated k via the curvature information.

Experiments are provided to compare the proposed method with the vanilla k-NN, on various data sets. Not surprisingly, this improves the performance over the baseline.

**Audience:**

Yes

**Broader Impact Concerns:**

None.

**Claims And Evidence:**

Yes

**Requested Changes:**

* I’m not sure if k-NN is really non-parametric, since it requires to specify the parameter k. At least this is not clear from your definition. You said “with a finite number of parameters remaining constant regardless of the size of the data set” to capture the parametric classifiers, and the parameter k in k-NN indeed seems like a parameter like this?

* Last paragraph of page 2, what is k-NNG? This is not defined (at this point).

* Last paragraph, third last line of page 2, the “log” in “k = log_2 n” is not in math font.

* It is unclear from the intro, that whether the curvature was considered in k-NN before. Is the use of curvature new in this paper?

* The set notation {\vec{x}_i \cup \eta_i} two lines above (1) does not make sense to me. You said \eta_i is the neighbors of \vec{x}_i, so I suppose it is a set of points? Then \vec{x}_i \cup \eta_i is a union of a point to a point set, which has a type inconsistency. Moreover, you take this union as another set, and this is even weird. I guess the correct notation is \eta_i \cup {\vec{x}_i}?

* What is “first fundamental form” and “second fundamental form”? How are these terms defined, or at least give a reference that discusses these? In fact, I don’t see how these are related to the discussion of \Sigma_i.

* In general, I find the description in the second half of page 5 not understandable. It seems to assume much knowledge from previous work. Another issue is that the intuition is explained well; for instance, “Thus, our approach is based in the strategy adopted by the manifold learning algorithm…” — why “Thus”? “The main problem is that the columns of X_i are not orthogonal” — why is this a “problem”? A “problem” for what?

* Is k = log_2 n in the first stage justified? Why not other values?

* It seems the time complexity of testing/prediction procedure is not analyzed.

* Looking at your testing procedure in Algorithm 3 — does it mean that your algorithm works the best for batch prediction? What if the query is submitted one at a time?

* In the experiments, how is k set for the vanilla k-NN method? I didn’t find this discussed.

* The current experiments only compare with the vanilla k-NN. I think this is not sufficient/convincing. In particular, there were previous works that also aim to find fine-tuned k (or other methods that aim to improve the performance of k-NN in general), and they should be compared.

**Strengths And Weaknesses:**

## Strength:

The result improves over the baseline. The overall idea is intuitive, and the implementation of the idea seems natural to me.

## Weakness:

* The experiments are not convincing (see my detailed comments)

* The complexity of the method seems very large. Even for the prediction/testing phase, the complexity is quadratic in the size of the testing dataset. The efficiency is not reported in the experiment, so it might be helpful to include one to justify this in the next revision.

* The writing quality should be improved (also see my detailed comments)

---

### Decision · Action_Editor_xoPm · 2024-06-25

**Recommendation:** Reject

**Comment:**

There is a consensus among reviewers to reject the paper in its current form. I would encourage the authors to carefully revise their manuscript and take into account the many remarks and suggestions from reviewers.

**Audience:**

Reviewers agree the work is relevant to TMLR's audience.

**Claims And Evidence:**

Empirical evidence appears weak.

**Resubmission Of Major Revision:**

The authors may consider submitting a major revision at a later time.